

# On the Appropriate Definition of Soil Profile Configuration and Initial Conditions for Land Surface-Hydrology Models in Cold Regions

Gonzalo Sapriza-Azuri[1], Pablo Gamazo[1], Saman Razavi[2,3,4] and Howard S. Wheater[2,3,4]

[1]Departamento del Agua, Centro Universitario Regional Litoral Norte, Universidad de la República, Salto, Uruguay
[2]Global Institute for Water Security, University of Saskatchewan, Saskatoon, SK, Canada
[3]School of Environment and Sustainability, University of Saskatchewan, Saskatoon, SK, Canada
[4]Department of Civil and Geological Engineering, University of Saskatchewan, Saskatoon, Saskatoon, SK, Canada

*Correspondence to*: Gonzalo Sapriza-Azuri (gsapriza@gmail.com)

**Abstract.** Arctic and sub-arctic regions are amongst the most susceptible regions on Earth to global warming and climate change. Understanding and predicting the impact of climate change in these regions require a proper process representation of the interactions between climate, the carbon cycle, and hydrology in Earth system models. This study focuses on Land Surface Models (LSMs) that represent the lower boundary condition of General Circulation Models (GCMs) and Regional Climate Models (RCMs), which simulate climate change evolution at the global and regional scales, respectively. LSMs typically utilize a standard soil configuration with a depth of no more than 4 meters, whereas for cold, permafrost regions, field experiments show that attention to deep soil profiles is needed to understand and close the water and energy balances, which are tightly coupled through the phase change. To address this, we design and run a series of model experiments with a one-dimensional LSM, called CLASS (Canadian Land Surface Scheme), as embedded in the MESH (Modélisation Environmentale Communautaire – Surface and Hydrology) modelling system, to (1) characterize the effect of soil profile depth under different climate conditions and in the presence of parameter uncertainty, and (2) develop a methodology for temperature profile initialization in permafrost regions, where the system has an extended memory, by the use of paleo-records and bootstrapping. Our study area is in Norman Wells, Northwest Territories of Canada, where measurements of soil temperature profiles and historical reconstructed climate data are available. Our results demonstrate that the adequate depth of soil profile in an LSM varies for warmer and colder conditions and is sensitive to model parameters and the uncertainty around them. In general, however, we show that a minimum of 20 meters of soil profile is essential to adequately represent the temperature dynamics. Our results also indicate the significance of model initialization in permafrost regions and our proposed spin-up method requires running the LSM over more than 300 years of reconstructed climate time series.



## 1 Introduction

Arctic and subarctic regions are amongst the most susceptible on Earth to climate change (IPCC 2013; Hinzman et al., 2005). For example, shrub expansion into the tundra regions (Sturm et al., 2001), permafrost thaw (Connon et al., 2014; Rowland et al., 2010), and glacier retreat (Marshall 2014) are some of the current manifestations of climate change. All these

changes are triggered by the interaction of climate, the carbon cycle and hydrology in response to global warming (Schuur et al., 2015). These effects are expected to be exacerbated due to global warming trends in the coming years (IPCC 2013; Slater and Lawrence 2013; Lawrence and Slater 2005). Therefore, being able to evaluate and assess the impact of climate change in cold regions is a primary concern for the scientific community, stakeholders and First Nations communities in northern regions. The significance of this problem in Canada has led to the creation of the Changing Cold Regions Network (DeBeer

et al., 2015; www.ccrnetwork.ca), which aims to provide improved science and modelling to address these concerns.

Earth system models are essential tools for evaluating the impacts of climate change. At global and regional scales, General Circulation Models (GCMs) and Regional Climate Models (RCMs) are used to simulate climate change evolution. Land Surface Models (LSMs) are used with GCMs and RCMs (coupled or offline) to represent the hydrological processes associated with the lower boundary condition of the atmosphere. These models typically represent the coupled energy and

water balance in the soil, based on numerical solution of the Richards' equation and using a relatively coarse vertical discretization.

In general, a standard soil configuration with a depth of no more than 4 meters is used in all LSMs that are commonly implemented in GCMs and RCMs (see for example the comparison made by Slater and Lawrence (2013) for the soil configuration depth in LSMs implemented in some GCMs). The typical boundary conditions to solve the energy and water

balance in the soil column are: (1) the exchanges with atmosphere at the top, (2) no lateral exchange of water or energy with the surrounding grids (only vertical fluxes), and (3) no heat flux at the bottom of the soil.

For moderate climate conditions and at the spatial scale on which these models are commonly applied, the above depth and boundary conditions are sufficient to capture the intra-annual variability in the energy and water balance. However, for cold regions, where the energy balance is closely related to the water balance through the phase change (Woo 2012), deeper soil

configurations and more representative boundary conditions are needed. A deeper soil profile allows the heat signal to propagate through the soil to deeper layers and hence avoids erroneous near-surface states and fluxes, such as overheating or over-freezing during summer and winter respectively (e.g: Lawrence et al., 2008; Stevens et al., 2007). Deeper soil/rock configurations, however, have longer system memories, and as such, particular care should be taken to define the initial conditions for the subsurface system. An alternative to modelling a deeper soil profile is the incorporation of a geothermal

heat flux as the lower boundary condition to the soil (Hayashi et al., 2007). However, in practice, the geothermal heat flux is usually not included in models due to lack of data.

The aforementioned challenges and shortcomings have been recognized by the climate, permafrost, and hydrology community. For climate models, Slater and Lawrence (2013), Alexeev et al. (2007), Nicolsky et al. (2007) and Stevens et al.




(2007), have disputed the validity of GCM future projections due to the shallow soil profile depth in LSMs for the reasons stated above. There are however examples of how the spatial distribution of permafrost is improved by including deeper soil configurations in a LSM. For example, Paquin and Sushama (2015), applied the Canadian RCM, which uses Canadian Land Surface Scheme (CLASS) (Verseghy, 1991) as the LSM, for the arctic region, and by considering a 65 m deep soil

configuration with a spin-up period of 200 years through recycling the 1970-1999 period, they improved spatial distribution of permafrost in cold regions. Zhang et al. (2003, 2006, and 2008) used a thermal soil model that includes soil water balance and showed the importance of considering deep soil configurations. Ednie et al. (2008) illustrated the necessity of a suitable model initialization to properly simulate soil thermal profiles in permafrost regions.

In the context of LSMs, Troy et al. (2012), simulated river basins in northern Eurasia using a 50 meter deep soil

configuration with a spin-up of 500 years by recycling the 1901-2001 period 5 times. Decharme et al. (2013), who applied the ISBA model to the whole of France, concluded that an 18 m depth was enough to properly simulate the energy and the water balance.

At a plot scale, Quinton et al., (2009, 2011) showed the importance of permafrost thaw in the hydrological model response. Hayashi et al., (2007) also showed the importance of incorporating adequate lower boundary conditions to simulate the

propagation of heat coupled with water flow in soils.

In light of the above, there is no doubt that deeper soil configurations in LSMs must be considered for simulating the land-surface hydrology in cold regions.  In addition, an increase in the soil configuration depth (SCD) results in a modelling system with longer memory, requiring longer spin-up periods for initialization. The presence of significant non-stationarity in climate and hydrology (Razavi et al. 2015) further challenges the process of model initialization and necessitates the

availability of long historical records in order to include past non-stationarity that may affect the present state and flux variables. Due to this non-stationarity, it may be inadvisable to initialize a model by recycling the historical records (i.e., repeating the simulation over the same period multiple times and using the final model state of one run as the initial state of the next run), as implemented in Troy et al., (2012) or Paquin and Sushama, (2015), since there is a warming trend in temperature which results in warmer and warmer soil states after each cycle.

Notwithstanding these facts, the depth considered and the way that initialization is set up in the literature are in general arbitrary. Moreover, the effect of model parameter uncertainty has not been considered in previous work, and only soil types from "look-up tables" and peat soils were compared (Paquin and Sushama 2015). The effect of the climate conditions used to spin-up has also not been analysed. The modeller often faces challenging questions, such as: (1) Do we have to set soil depth to 20 m, 30 m or 60 m? (2) If we use 30 meters, do we need to spin-up over 150, 500, or 1000 years? Do we have to

use a sequence of years with different hydroclimatic conditions or one year with a particular condition? Or to go further, do we have to simulate longer historical periods by generating synthetic climatic time series based on proxy records such as tree-ring widths? What are the effects of model parameters and the uncertainty around them in the definition of the model configuration? This study is an attempt to address these questions.





## 2 Methods

To advance our understanding and modelling capability of soil moisture and energy dynamics in permafrost regions, we developed two numerical experiments for a study area located in the Northwest Territories, Canada, where observations of soil temperature at several depths and historical reconstructed climate data are available.

### 2.1 Study Area and Data

The experimental test case is located at Norman Wells, in the Mackenzie Valley, Northwest Territories, Canada (Figure 1). Based on the Permafrost Map of Canada (Geological Survey of Canada, 2000), the area is located in a zone of extensive discontinuous permafrost. The main land cover is characterized by grass and the subsurface is formed by ice-rich silt clays. The climate of the region is subarctic, according to the Köppen climate classification (Pell et al., 2007), with an average

annual mean daily temperature of -5 ℃ and average annual precipitation of 295 mm/year.

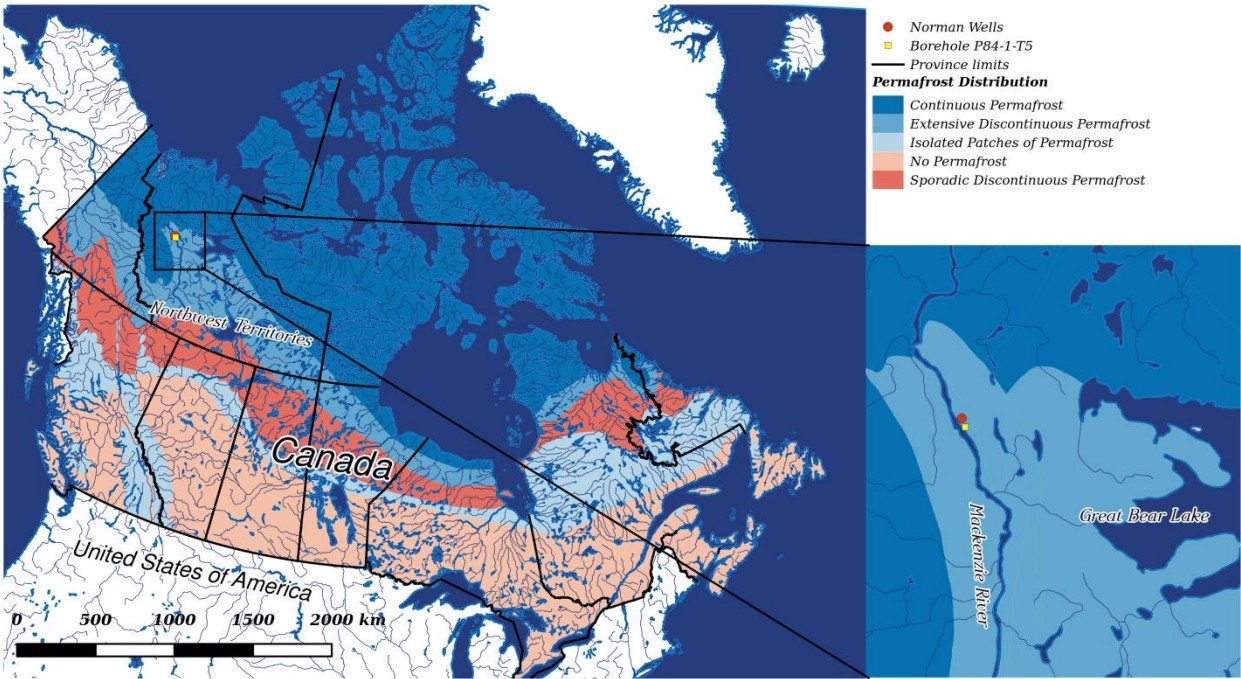

**Figure 1: Permafrost Map of Canada and location of the area of study. Temperature soil profiles are available at the borehole P84-1-T5 (yellow dot).**

This area is selected due to the availability of both soil temperature at several depths down to 20 meters (Smith et al., 2004) and dendroclimatic reconstructions of summer air temperature (Szeicz and MacDonald, 1995). These data will be used to test the proposed methodology to define the SCD and the initialization approach.



### 2.1.1 Soil Temperature Profiles

Administrated by the Geological Survey of Canada (Smith et al., 2004), annual soil temperature profiles are available based on the maximum and minimum daily average of soil temperature at several borehole locations in the Mackenzie Valley. Figure 2 shows the temperature profiles for the borehole 84-1-T5 selected for our analysis. The soil temperatures were

5 measured at the following depths (in meters) {1.0, 2.0, 3.0, 4.0, 6.0, 8.0, 10.0, 12.0, 15.0, 18.0, 19.6} for period 1985-2001. The active layer thickness, defined as the soil depth that encapsulates the seasonal freeze-and-thaw cycle (Woo, 2012) , was also reported and varied from 1.5 m at the beginning of the period of records (1985) up to 3.0 m to the end of the period (2000), showing an increasing trend in the active layer thickness over time.

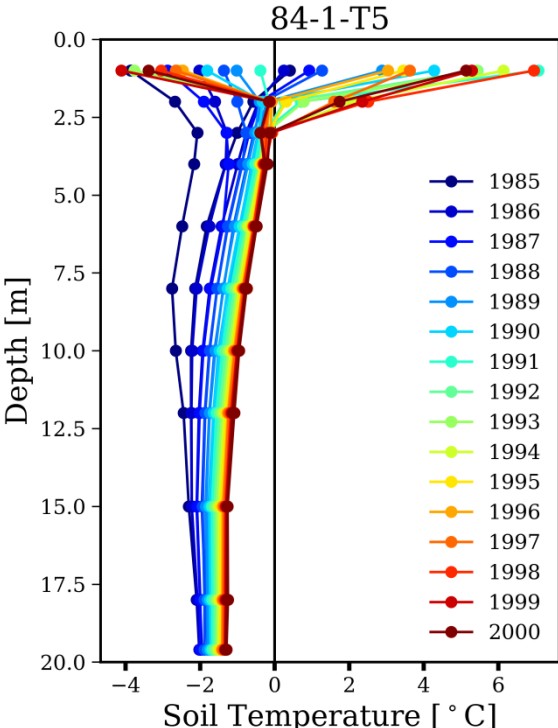

**Figure 2: Permafrost Annual maximum and minimum soil temperature profiles for the borehole 84-1-T5 located in Normal Wells. Each colour represent an individual year (1985-2000).**

### 2.1.2 Reconstructed Summer Air Temperature

Szeicz and MacDonald (1995) generated proxy climate records of average summer (June-July) air temperature based on tree rings for period 1638-1988 in north-western Canada near to Norman Wells (Figure 3). These proxy data have been

previously used by other authors (Edine et al. 2008; Esper et al., 2002). For example, Edine et al. (2008) showed that the linear trend of proxy summer air temperature can be used as an approximation of the linear trend of the mean annual air




temperature for the region. Following this approach, we generate a stochastic climate time series (Section 3.5) that follows the historical reconstructions of mean annual air temperature based on the proxy data of Szeicz and MacDonald (1995).

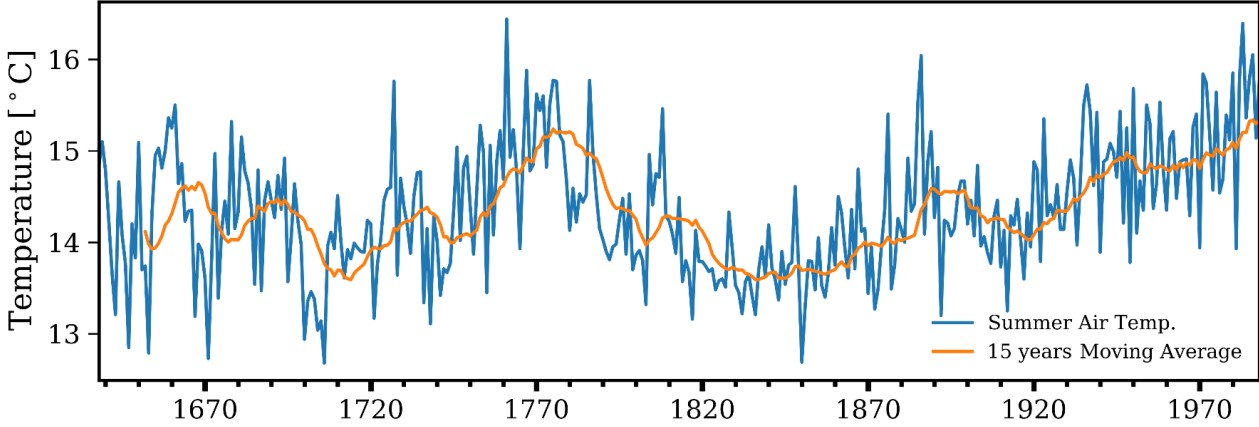

**Figure 3: Reconstructed summer (June-July) air temperature from age-dependent tree ring modelling for period 1638-1988 with a 15 year moving average.**

## 2.2 Design of Experiments

The methodology and experiments were designed to be carried out in two stages. In the first stage, we focus on the characterization of the adequate soil profile depth for land surface-hydrologic modelling in the permafrost regions, in relation to climate condition and model parameterization. For this purpose, we run a 1D model under a variety of soil profile, parameter, and climate configurations. This stage is referred to as "Experiment 1" in this paper.

In the second stage, we propose a method to handle the presence of non-stationarity in climate and hydrology, in order to include effects of past non-stationarity on the present state and flux variables. This method utilizes paleo-climate reconstructions to generate long, synthetic time series of climate variables for model initialization. We call this stage "Back to the past".

## 2.3 The 1D-Model

The core of the experiments is a 1D model implemented in MESH, Environment and Climate Change Canada's community model (Pietronero et al., 2007). This integrates the CLASS LSM (Verseghy et al., 1993; Verseghy 1991), which solves coupled energy and water balance equations for vegetation, snow and soil and their exchange of heat and moisture with the atmosphere, and WATROF (Soulis et al., 2000) or PDMROF (Mekonnen et al., 2014) to solve the horizontal flow processes for basin-scale integration. MESH discretizes the spatial domain based on regular grid cells and each individual cell is then subdivided in Grouped Response Units (GRUs) based on land cover and/or soil types. The 1D CLASS model is implemented here at a point, and a unique GRU based on grass land cover was used. The upper boundary condition of the



model is formed by atmospheric forcings. No heat flux is assumed as the lower boundary condition, and the water flux that reaches the bottom of soil profile drains to generate base flow.

The climate forcings needed are temperature, precipitation, shortwave radiation, longwave radiation, specific humidity, wind velocity and atmospheric pressure.

## 5 2.4 Experiment 1

A schematic representation of the model experiment is illustrated in Figure 4. Several 1D model set-ups were implemented by a combination of (1) various soil depth configuration, (2) several climate conditions selected to spin-up the model and (3) different values for the parameters that control hydrological processes (water and energy balance).

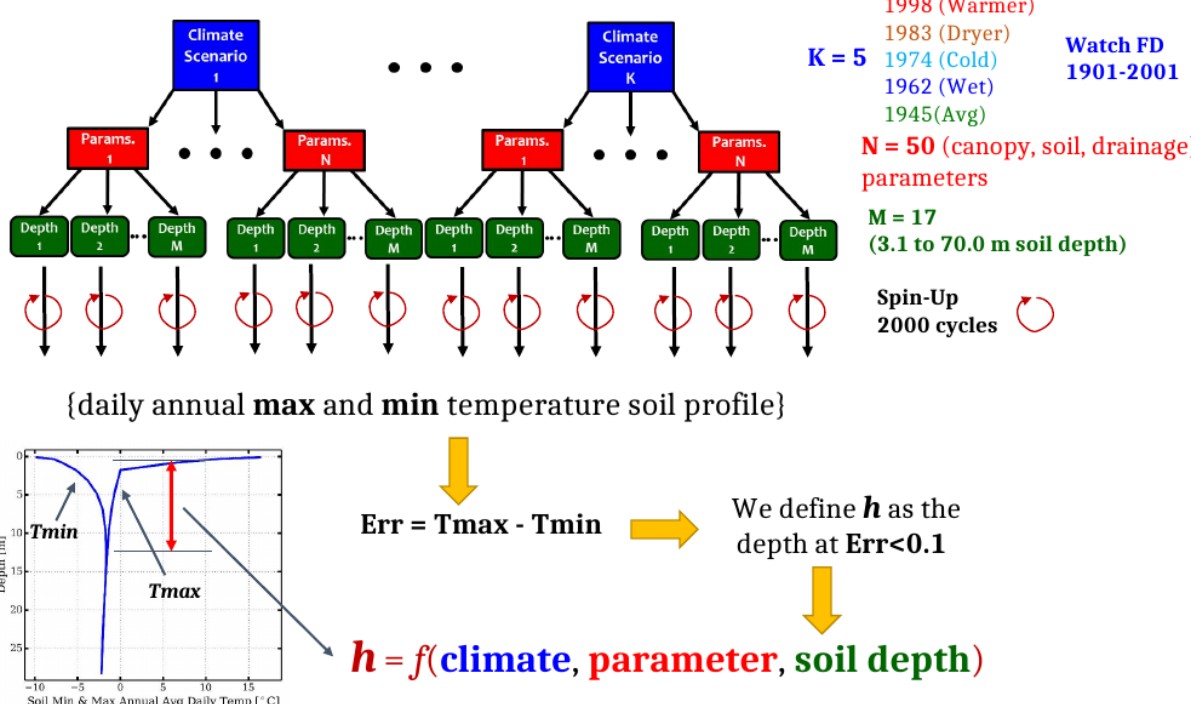

**10 Figure 4: Schematic representation of the model experiment for Experiment 1. The model set-ups are defined as combinations of 5 different climate condition, 50 sampling parameters and 17 different soil configurations. Each model is then run in a spin-up mode for 2000 cycles. The last year of spin-up is taken to compute the daily annual max and min soil temperature profile and their difference is computed. At the depth where such difference is less than 0.1 we define the depth as the $h_T$-non-oscillation condition.**

### 2.4.1 Variable Soil Depth Configuration

15 For the experiment, a series of 1D models with an incremental number of soil layers (different total soil depths) are defined. The soil configurations of the 1D models are specified in Table 1, and range from the standard CLASS configuration of 3 layers with a 4.1 meter depth up to 20 layers corresponding to a depth of 71.59 m. The width of each layer is increased exponentially. A total of 17 different soil configurations are tested.


| Soil Config | Nº Soil Layers | Depth of each layer [m] | Total Depth [m] |
|---|---|---|---|
| 1 | 3 | 0.1 , 0.25, 3.75 | 4.1 |
| 2 | 5 | 0.1, 0.25, 0.51, 0.91, 1.32 | 3.09 |
| 3 | 6 | 0.1, 0.25, 0.51, 0.91, 1.32, 1.72 | 4.81 |
| 4 | 7 | 0.1, 0.25, 0.51, 0.91, 1.32, 1.72, 2.13 | 6.94 |
| 5 | 8 | 0.1, 0.25, 0.51, 0.91, 1.32, 1.72, 2.13, 2.54 | 9.48 |
| 6 | 9 | 0.1, 0.25, 0.51, 0.91, 1.32, 1.72, 2.13, 2.54, 2.94 | 12.42 |
| 7 | 10 | 0.1, 0.25, 0.51, 0.91, 1.32, 1.72, 2.13, 2.54, 2.94, 3.35 | 15.77 |
| 8 | 11 | 0.1, 0.25, 0.51, 0.91, 1.32, 1.72, 2.13, 2.54, 2.94, 3.35, 3.75 | 19.52 |
| 9 | 12 | 0.1, 0.25, 0.51, 0.91, 1.72, 2.13, 2.54, 2.94, 3.35, 3.75, 4.16 | 23.68 |
| 10 | 13 | 0.1, 0.25, 0.51, 0.91, 1.32, 1.72, 2.13, 2.54, 2.94, 3.35, 3.75, 4.16, 4.57 | 28.25 |
| 11 | 14 | 0.1, 0.25, 0.51, 0.91, 1.32, 1.72, 2.13, 2.54, 2.94, 3.35, 3.75, 4.16, 4.57, 4.97 | 33.22 |
| 12 | 15 | 0.1, 0.25, 0.51, 0.91, 1.32, 1.72, 2.13, 2.54, 2.94, 3.35, 3.75, 4.16, 4.57, 4.97, 5.38 | 38.60 |
| 13 | 16 | 0.1, 0.25, 0.51, 0.91, 1.32, 1.72, 2.13, 2.54, 2.94, 3.35, 3.75, 4.16, 4.57, 4.97, 5.38, 5.79 | 44.39 |
| 14 | 17 | 0.1, 0.25, 0.51, 0.91, 1.32, 1.72, 2.13, 2.54, 2.94, 3.35, 3.75, 4.16, 4.57, 4.97, 5.38, 5.79, 6.19 | 50.58 |
| 15 | 18 | 0.1, 0.25, 0.51, 0.91, 1.32, 1.72, 2.13, 2.54, 2.94, 3.35, 3.75, 4.16, 4.57, 4.97, 5.38, 5.79, 6.19, 6.60 | 57.18 |
| 16 | 19 | 0.1, 0.25, 0.51, 0.91, 1.32, 1.72, 2.13, 2.54, 2.94, 3.35, 3.75, 4.16, 4.57, 4.97, 5.38, 5.79, 6.19, 6.60, 7.00 | 64.18 |
| 17 | 20 | 0.1, 0.25, 0.51, 0.91, 1.32, 1.72, 2.13, 2.54, 2.94, 3.35, 3.75, 4.16, 4.57, 4.97, 5.38, 5.79, 6.19, 6.60, 7.00, 7.41 | 71.59 |

**Table 1: The variable soil configuration profiles defined for the 1D model: number of soil layers, depth of each layer and total depth. Each colour in column 3 represent grouped layers and assigned the same values to the parameters of the layers in each group except for the first soil configuration and the first two layer for all the soil configurations (black colour).**

## 2.4.2 Climate Conditions

To account for the effect of climate conditions, years 1998 (warm), 1983 (dry), 1974 (cold), 1962 (wet), and 1945 (average) are used with each model configuration. Each model was run over five 2000-year-long sequences, each of which comprised 2000 back-to-back repetitions of one of the above years. These five climate conditions are defined based on temperature and precipitation obtained from the WATCH FD (WCH-FD) gridded data base of climate forcing (Weedon et al., 2011) for the period 1901-2001 at the location of our study area. We do not use the historical sequence of years 1901-2001 to avoid overheating effects that could be introduced due to the warming trend of the last past century.

## 2.4.3 Parameters

Three groups of parameters representing canopy, soil and drainage processes are perturbed to analyze their influence on SCD. Table 2 describes all the parameters considered along with their lower and upper intervals of variation. Monte Carlo sampling with a uniform distribution is applied to generate a collection of 50 samples for each parameter. To set a consistent parametrization scheme for the soil texture across the models with different numbers of layers, we grouped layers and assigned the same values to the parameters of the layers in each group. These groups are represented with different colors in Table 1, column 3 (Depth of each layer).



| Id | Name | Units | Lower Bound | Upper Bound | Description |
|----|------|-------|-------------|-------------|-------------|
| 1 | LAMX | [-] | 2.0 | 4.0 | Annual Max leaf-area index |
| 2 | LAMN | [-] | 2.0 | 4.0 | Annual Min leaf-area index |
| 3 | ALVC | [-] | 0.03 | 0.06 | Avg visible albedo of the vegetation when fully-leafed |
| 4 | ALIC | [-] | 0.2 | 0.34 | Avg near-infrared albedo of the vegetation when fully-leafed |
| 5 | ROOT | [m] | 0.2 | 1.55 | Root depth |
| 6 | SDEP | [m] | 2.0 | Max Depth | Permeable Depth |
| 7 | GRKF | [-] | 0.001 | 1.0 | Fraction of the saturated surface soil conductivity moving in the horizonal direction |
| 8 | KSAT | [m s-1] | 0.0001 | 5.5 | Saturated surface soil hydraulic conductivity |
| 9 | SAND* | [%] | 0.0 | 100 | % sand texture |
| 10 | CLAY * | [%] | 0.0 | 100 | % clay texture |
| 11 | ORG * | [%] | 0.0 | 100 | % material organic texture |
| 12 | ZSNL | [m] | 0.05 | 0.5 | Minimum depth to consider 100% cover of snow on the ground surface |
| 13 | ZPLS | [m] | 0.05 | 0.5 | Maximum depth of liquid water allowed to be stored on the ground surface for snow-covered areas |
| 14 | ZPLG | [m] | 0.05 | 0.5 | Maximum depth of liquid water allowed to be stored on the ground surface for snow-free areas |

**Table 2: Parameters list description with the upper and lower bound interval used. For the texture parameter (*) SAND, CLAY and ORG the sampling is made to sum to 100%.**

### 2.3.4 Non-Oscillation Depth

In Experiment 1, we ran a total of {(17 SCD)*(5 climates)*(50 parameters)}= 4250 model combinations. In each of these model set-ups, a 2000-year model run was performed. All the models were set with the same initial conditions and constant temperature and liquid/ice saturation soil profiles. The soil thermal profile was defined at -3.0 °C and all the soil water was defined as ice content. We assume that after the spin-up a quasi-equilibrium between the climate conditions and the ground thermal state was reached. The last cycle, a complete 1 year simulation, was used to compute the annual soil temperature

profiles based on the maximum (maxTsp) and minimum (minTsp) daily average of soil temperature (Figure 4). Next, we compute the difference between maxTsp and minTsp and define a depth (h) where this difference was less than 0.1 °C. We name this depth h as the "non-oscillation depth" of annual soil temperature. Therefore, h, which is a function of climate condition, parameter values, and simulated soil depth, represents the depth at which the soil thermal response remains invariant over a season. In other words, the non-oscillation depth indicates the depth at which the SCD has not longer a

significant effect on the energy balance computed by the model.



## 2.5 Experiment 2: Back to the past

To be able to simulate the hydrology using LSMs in cold regions in the last century (period of records) and in the future, it is necessary to correctly set the initial conditions of the models. When the SCD of the model is considered to be shallow (no more than 4 meters), the initialization can be easily carried out with a relatively short spin-up period. However, with deeper

SCDs, the memory of the system is longer, and it remembers the past climate regimes and trends. Therefore, it is necessary to run the model over an extended period of time to diminish the effect of uncertainty in initial conditions on model predictions. This is a major challenge, however, as the typical length of periods of records (say ~100 years) is not sufficient. To overcome this challenge, we stochastically generated past climate variables, back to year 1678 based on proxy data of reconstructed summer air temperature described in section 2.1.2. To this end, we applied a block bootstrapping technique

(Razavi et al., 2015; Politis and Romano 1994).

The stochastic time series of climate variables were generated as follows:

(1)  First, we assumed that the reconstructed summer air temperature by Szeicz and MacDonald (1995) can be used as proxy data to derive the past trends in air temperature. The historical temperature trend back to 1678 ($T_{Htrend}$) was estimated by first computing the moving average with a window of 15 years and then subtracting

the moving average from the annual time series. Figure 5 compares both temperature trends (15-year moving average) obtained from WCH-FD data and tree ring for the same period showing good agreement, with a Pearson correlation coefficient of 0.66. The existing discrepancy may be in part due to a lack of consideration of longer-term variability (longer than annual) in the reconstruction of the time series, an issue explained in Razavi et al., (2016).

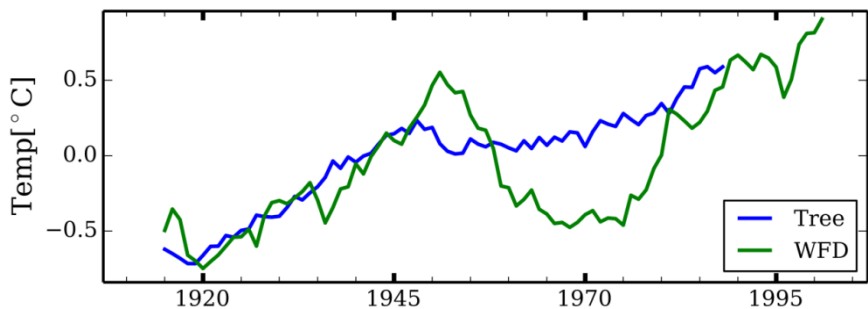

**Figure 5: Trend comparison of annual average air temperature with subtracted mean for the whole period for WCH-FD and tree ring data.**

(2)  Then, we decomposed the WCH-FD temperature time series (6 hour time resolution) for the period 1901-2001 into its trend and its seasonality component ($T_{seas}$).

(3)  Next, we applied the block bootstrapping technique with a block size of 5 years to $T_{seas}$. We sampled 45 blocks of 5 years so as to generate a time series long enough to cover the 1678-1901 period.



(4)    To finish the reconstructions of the 6-hourly time resolution of temperature we added $T_{seas}$ to the $T_{Htrend}$ from step (1).

(5)    The other six climate variables needed by MESH to run were precipitation, shortwave and longwave radiation, specific humidity, wind, and atmospheric pressure. They were generated by applying the block bootstrapping with the same time indexes of the temperature blocks (step 3). In this way, we maintained the interdependence between all the climate variables.

(6)    Finally, we generated 100 realizations of the climate variables for period 1678-1901. The complete climate time series of 1678-2000 was finally obtained by combining the generated ones and the WCH-FD data for 1901-2000. Figure 6 shows the mean annual temperature generated with the methodology presented.

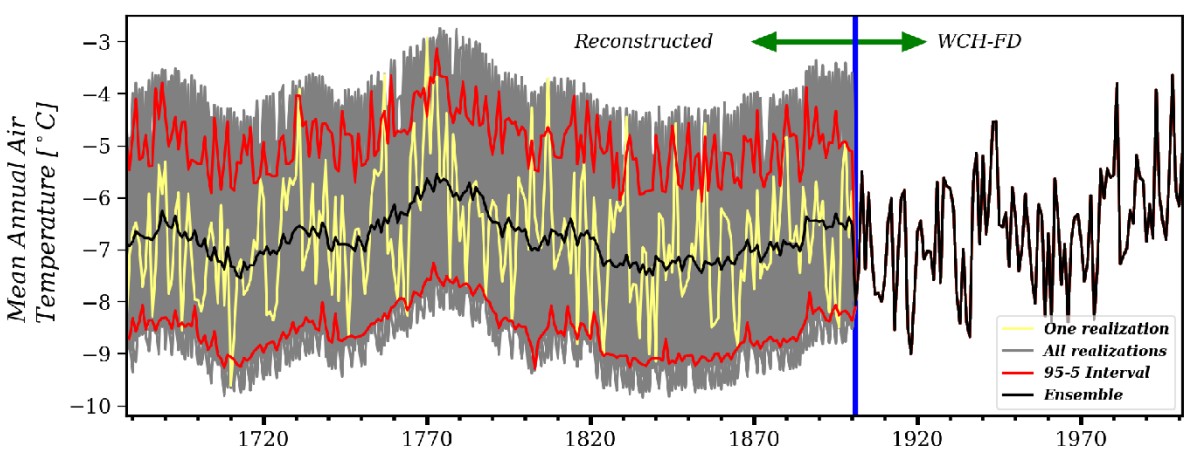

**Figure 6: Combined air Temperature time series generated using the block bootstrapping technique and WCH-FD. The time series is divided in two periods. From 1678-1900 the temperature and the other 6 climate variables are generated using the block bootstrapping with a block of 5 years. In the figure are shown 100 realizations (grey lines), the 95-5 % confident interval (red lines) and the average of the ensemble (black line). The second period (1901-2000) the climate variable are used directly from the WCH-FD.**

### 2.5.1 Evaluation procedure

We used the 100 realizations of the climate variables of section 2.5 to run the models with the 100 parameter sets and 17 SCDs used before. For the initial conditions, we used the stabilized model outputs obtained from the 2000 cycles for the year 1945 (average with respect to temperature and precipitation). Finally, the simulated soil temperature profiles obtained were compared with the observed (see section 3.1.1) by computing the root mean square error (RMSE) to evaluate if it is possible to reproduce the soil thermal behaviour. The RMSE was computed by calculating, for each individual simulation of the annual soil temperature profile, the annual minimum and maximum daily soil temperature at the same location as that at which the observed soil temperature was measured (section 3.1.1). To have a more general view of the model performance in reproducing the observations (1985-2000), individual maximum and minimum soil temperature profile of simulated and





observed were used to compute a RMSE for each individual year. Then all the values of RMSE obtained (maximum and minimum) for each year were averaged to obtain a unique RMSE.

## 3 Results

### 3.1 Soil Configuration Depth

Using the experiments proposed in Experiment 1, we explored the combined and individual effects of climate, parameters and SCD on the non-oscillation depth of annual soil temperature. Figures 7, 8 and 9 summarize these analyses as 2D histograms: (SCD, $h_T$-non-oscillation) (Figure 7); (years, $h_T$-non-oscillation) (Figure 8); and (parameter sample group, $h_T$-non-oscillation) (Figure 9). Notably, Figure 7 shows that for SCDs less than 15 m, there is a high probability that the $h_T$-non-oscillation condition is never reached, independently of the parameter value selection and the climate conditions (year).

For SCDs of greater than 20 m, the $h_T$-non-oscillation condition is always reached with a higher probability that this condition occurs at a depth between 13 and 16 m.

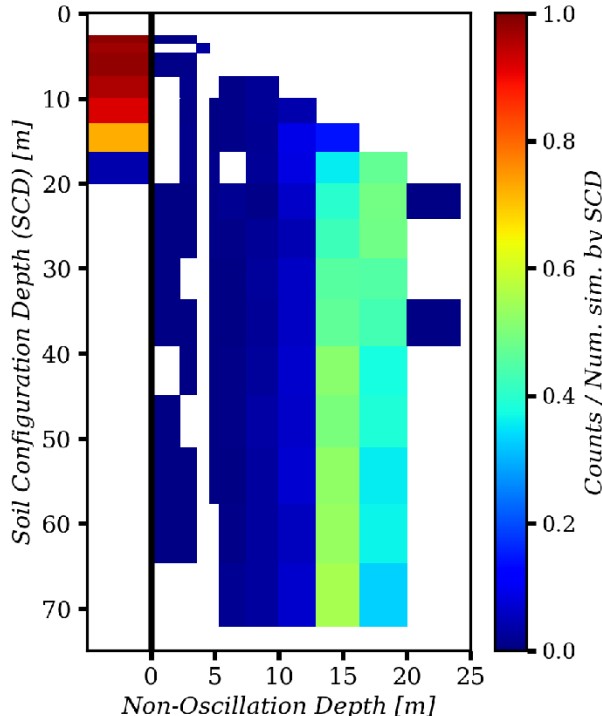

**Figure 7: 2d-Histogram of SCD and $h_T$-non-oscillation depth. Counts are normalized by the number of simulation by SCD. The black line represents the limit to reach or not the $h_T$-non-oscillation conditions. Bins to the left represent SCDs that never reach the**
**$h_T$-non-oscillation condition.**





The variability observed in ***h_T-non-oscillation*** depth for each SCD is, in general, mainly explained by the parameters rather than the year selected for the spin-up (Figure 8 and 9).

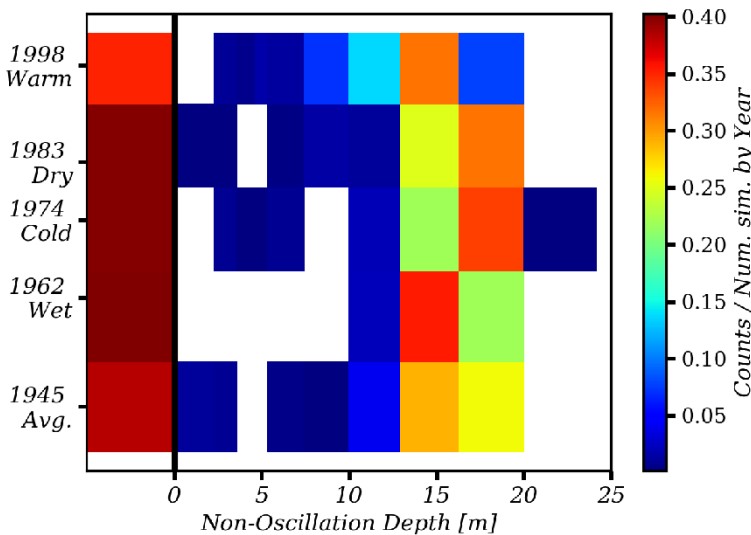

**Figure 8: 2d-Histogram of climate condition (years) and *h_T-non-oscillation* depth. Counts are normalized by the number of simulation by year. The black line represent limit to reach or not the *h_T-non-oscillation* conditions. Bins to the lefts represent 1d model that never reach the *h_T-non-oscillation* condition.**





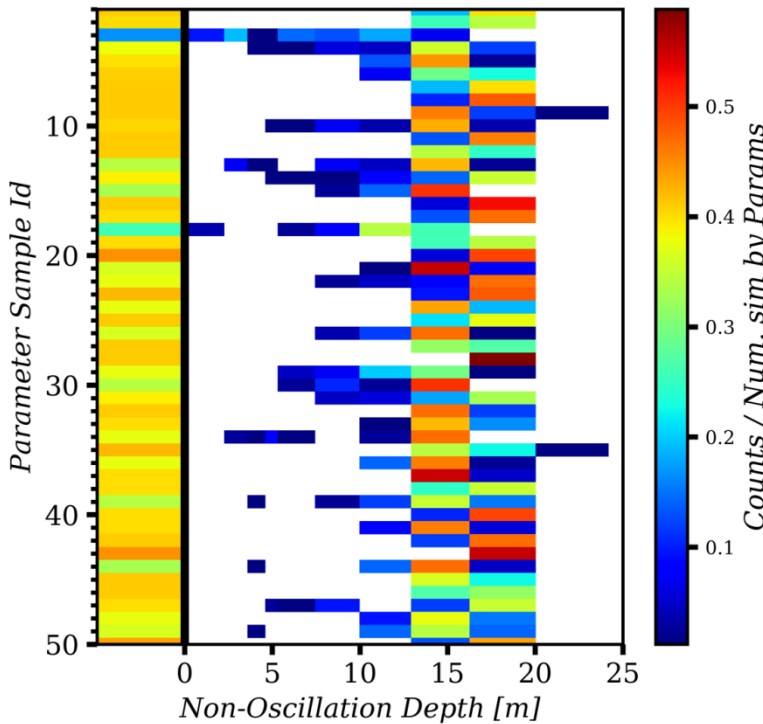

**Figure 9: 2d-Histogram of parameter and *$h_T$-non-oscillation* depth. Counts are normalized by the number of simulation by parameter sample. The black line represent limit to reach or not the *$h_T$-non-oscillation* conditions. Bins to the lefts represent 1d model that never reach the *$h_T$-non-oscillation* condition.**

From the previous results, it seems clear that we need at least an SCD of greater than 20 m to adequately represent the temperature dynamics of permafrost. This conclusion is supported by the fact that the soil temperature at which ***$h_T$-non-oscillation*** condition is reached remains invariant throughout the annual cycle. The distribution of this "non-oscillating temperature" is shown using 2d-histograms in Figure 10 and 11 with respect to the SCD and the climate conditions (years), respectively.

Figure 10 shows that for shallow SCD, from 3.1 m up to 16 m, there is a tendency to obtain a warmer soil temperature such that the permafrost is thawed. In the SCDs with the depth of 16 m and deeper, there is much more variability in the soil temperature (between -6 ℃ to 0 ℃), but with a high probability that the soil temperature at ***$h_T$-non-oscillation*** condition is between -3 ℃ to -2.5 ℃. In Figure 11 the effect of the climate condition can be appreciated. The main behaviour difference is for the warmest year (1998) when, as expected, the warmest soil temperatures at the ***$h_T$-non-oscillation*** condition occur.

As for the other climate conditions, the behaviours are quite similar and in general have a range of variation between -7℃ to 0.5 ℃. As before (Figure 10), the probability distribution for each climate condition is quite symmetrical with a peak value around -2.5℃. A slightly cooler soil temperature is obtained for the coldest year (1974).





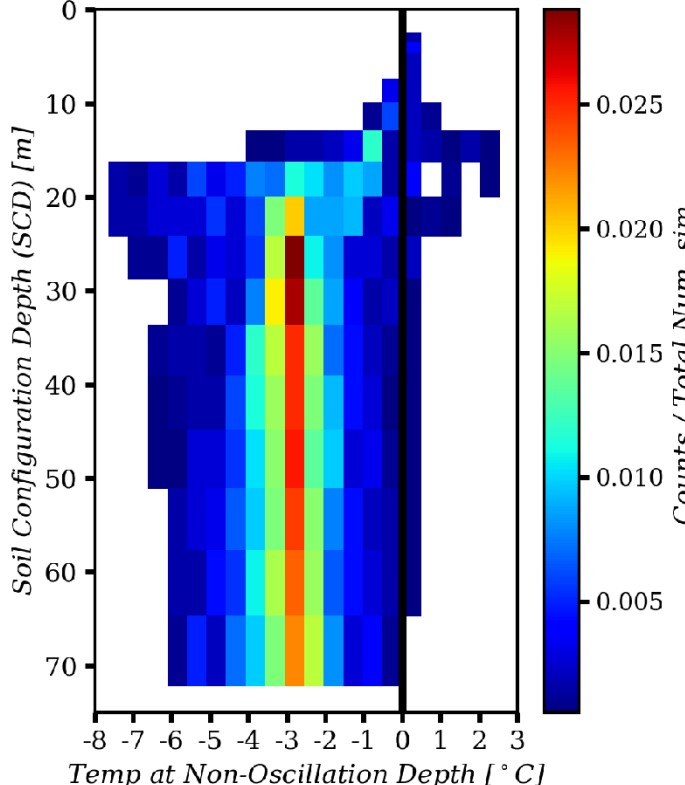

**Figure 10: 2d-Histogram of SCD and temperature at *h_T-non-oscillation* depth. Only SCD that have reached the *h_T-non-oscillation*
condition are included. The black line represent the 0 ºC temperature.**

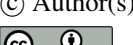


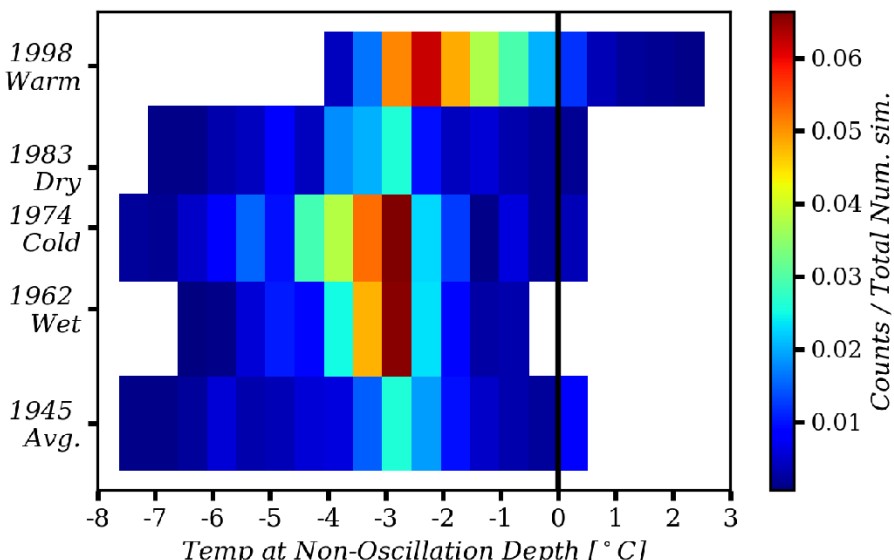

**Figure 11: 2d-Histogram of climate condition (years) and temperature at *$h_T$-non-oscillation* depth. Only SCD that have reached the *$h_T$-non-oscillation* condition are included. The black line represent the 0 ℃ temperature.**

### 3.2 Initialization by going Back to the Past

5  The previous section has shown evidence that regardless of the climate conditions and model parameters, we need to have an SCD that is deeper than 20 m. However, such depths make the model initialization problem more challenging. Here we show the results from driving our 1D model (varying SCD and parameters) applying a set of 100 reconstructed climate forcing realizations by going back to the past (1678-2001).

A general overview of the model's ability to reproduce (or not) the observed soil thermal behaviour between the year 1985 to

10  2000 is presented in Figure 12. We plot a 2d histogram that compares the SCD against the RMSE. The colours represent the probability of a RMSE value for a specific SCD that includes the effect of different parameter values and climate forcing realizations. The RMSE was calculated as described in section 2.5.1. In general, for the shallower SCD, the RMSE is larger with a higher variability (1.5 ℃ to 9.0 ℃). As deeper SCDs, the behaviour becomes quite uniform for all SCDs, with a range of RMSE between 1 ℃ to 5 ℃ and a higher probability that the RMSE is around 1.5 ℃ to 3.0 ℃.

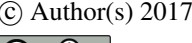


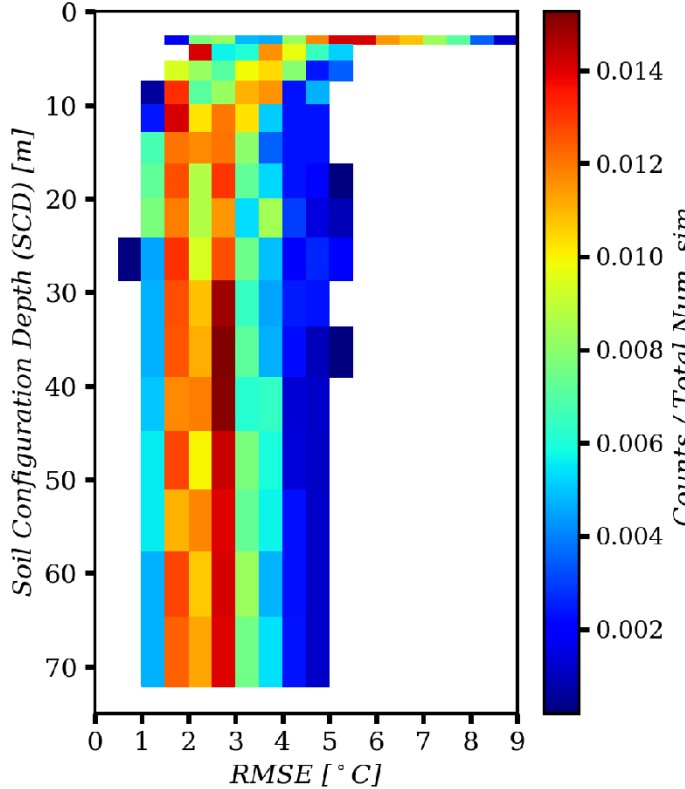

**Figure 12: 2d-histogram of SCD and RMSE. All the SCD are included regardless if the SCD have reached the *$h_T$-non-oscillation* condition.**

In the previous comparison (Figure 12), all the SCDs were included even if the soil temperature at the *$h_T$-non-oscillation* condition has not been reached for an individual SCD. In Figure 13 we compare only the SCDs that have reached the *$h_T$-non-oscillation* condition with the RMSE, in a 2d-histogram. For the SCDs that are deeper than 16.0 m, the behavior is quite similar to those obtained in Figure 12. This is explained by the fact that almost all the SCDs that are sufficiently deep (>16.0 m) reach the *$h_T$-non-oscillation* condition.





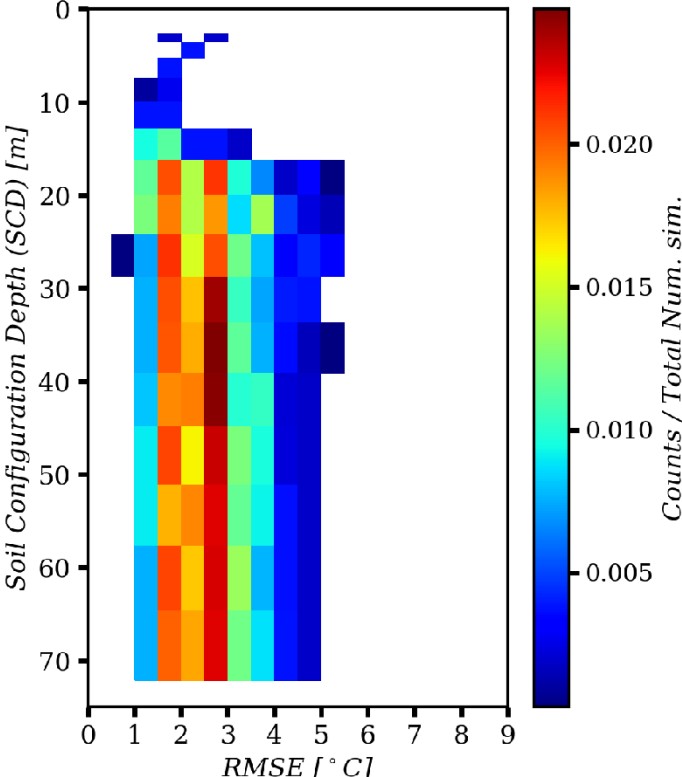

**Figure 13: 2d-histogram of SCD and RMSE. Only the SCD that reached the $h_T$-non-oscillation condition are included.**

To identify the relative effect of the different realizations of the reconstructed past climate and parameters on the variability obtained in RMSE, we plot a 2d-histogram comparing RMSE and parameter sample (Figure 14). Here, we are only taking into account the SCDs that have reached the **$h_T$-non-oscillation** condition. Results in Figure 14 show that for each parameter set the RMSE is narrow. Therefore, the variability obtained in RMSE in Figure 13 is mainly attributed to the parameter variability, and the effect of stochasticity in the reconstructed time series is minimal. This result reinforces the importance of adequately reproducing the long term trends in data used for model initialization.



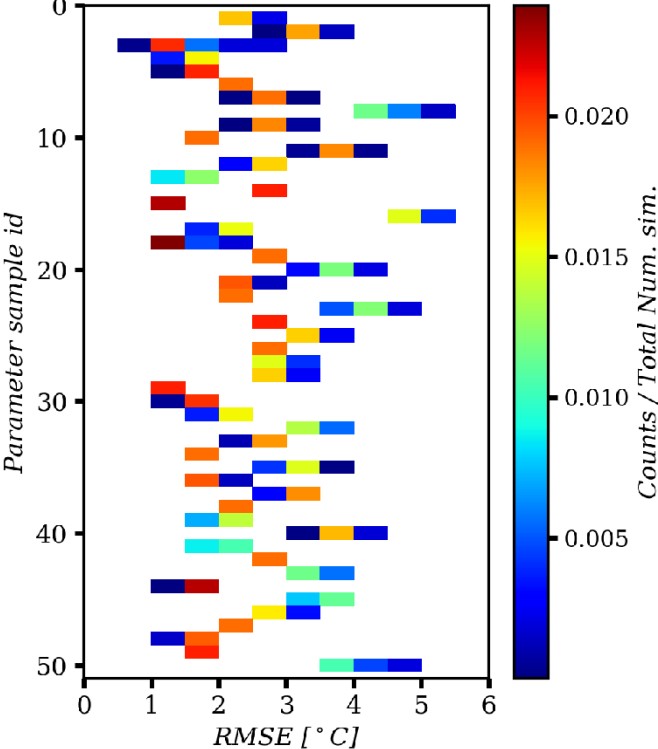

**Figure 14: 2d-Histogram of parameter sample and RMSE. Only the SCD that reached the *h_T-non-oscillation* condition are included.**

## 4 Discussion and conclusions

This study concludes that for permafrost regions, deeper soil configurations in LSMs are needed than commonly adopted, to be able to correctly simulate the coupled energy and water balance in the subsurface. This conclusion can be extended to all earth system models that incorporate a LSM and permafrost representation. While this conclusion has been pointed out by other authors, this work incorporates a rigorous analysis of the SCD, which evaluates the effects of parameter uncertainty, total soil depth, climate conditions and how the initialization should be carried out.

Our analysis shows that the minimum total soil depth should be around 20 m. This value is independent of the parameter selection and climate condition used to initialize the model. The metric defined to assess this depth was based on a depth at which the annual maximum and minimum of daily soil temperature are equal, referred to as *h_T-non-oscillation* condition in this paper. This depth represents a thermal stable condition and ensures that the lower boundary condition is deep enough to accommodate a no-heat-flux boundary condition at the bottom of the soil configuration. An alternative, not explored here, is

to consider a variable heat flux at the bottom boundary and reduce the total SCD.





The variability observed in the values of *hᴛ-non-oscillation* was mainly explained by the parameters rather than the climate conditions. This result is valid for both analyses: Experiment 1 and 'Back to the Past,' the long term simulation using stochastic reconstructed climate time series. This emphasizes the importance of recognizing and addressing parameter uncertainty and raises serious issues with the common practice in using LSMs with GCMs, where model capabilities are

constrained by using hard coded parameters determined based on look-up tables (Mendoza et al., 2015).

We argued that model spin-ups that are based on recycling of the 20th century data, or a sequence of years with trend should be avoided. Instead, to define the initial condition of the model, we recommend to proceed in two stages: Experiment 1 can be used to explore sensitivity of the soil depth and parameterisation and then "back to the past" to generate the relevant initial conditions. This should always be the case when deeper SCDs are going to be implemented in a LSM in cold regions.

The first stabilization assures that coherent state variables and fluxes are set before subsequent initialization of the model. This is an important step, as the majority of the LSMs have multiple variables and fluxes to initialize (e.g. CLASS has 17). For the first step, we recommend selecting an average year in term of air temperature and precipitation, and recycle that year up to the point that stabilization in soil temperature profile is reached. Then, in the second step, we recommend generating multi-century long records based on paleo-reconstructions, and running the model of step 1 on that. This will let the model

evolve over time on the time period preceding the period of records as to be able to simulate current conditions. Here, we reconstructed past climate using proxy data of summer temperatures derived from tree rings, applying block bootstrapping. We were able to reproduce quite well the past trends of summer temperature and we included the effect of uncertainty in the climate time series by generating 100 realizations. The number of years that are necessary to go back to the past will be a function of how deep is the SCD chosen. Deeper SCDs retain more memory of past climate and require longer spin-up

periods.

Finally, we envision our future work being directed to generalize the results obtained here by extending the analyses to other places where observations (of past climate and soil profile temperature) are available, increasing the number of parameters sampled to better explore the parameter space, and comparing several model parametrizations, including the effect of heat flux as a lower boundary condition at the bottom of the soil. For application in regional and global models, the SCD can be

variable as was proposed by Brunke et al. (2016) in the Community Land Model version 4.5. However, overall, computational burden is a bottleneck for large-scale simulations. To address the computational issues, an endeavour may be made by the cryosphere community to generate a unified gridded data set for the last millennium or so (1000 years back to the past) (Jungclaus et al., 2016; Landrum et al., 2013; Schmidt et al., 2011) that approximates soil temperature profiles with adequate soil depth and the effect of parameter uncertainty by considering different ensembles.





## Acknowledgment

This research was undertaken as part of the Changing Cold Region Network, funded by Canada's Natural Science and Engineering Research Council, and by the Canada Excellence Research Chair in Water security at the University of Saskatchewan.

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
