# Peer review of "On the Appropriate Definition of Soil Profile Configuration and Initial Conditions for Land Surface-Hydrology Models in Cold Regions"

_Hydrology and Earth System Sciences, 2017_

## Referee Comment (RC1) · Anonymous Referee #1 · 26 Dec 2017

GENERAL COMMENTS

The manuscript presents a sensitivity analysis of a Canadian one-dimensional land surface model, MESH to the thickness of modeled soil profile and the length of model initialization period. The main conclusions are that a soil profile of 20m or greater is necessary for this particular model to represent the energy dynamics of permafrost, and that an initialization period much longer than 100 years is necessary to condition the model properly. The same results have been reported by a number of previous researchers using different permafrost models, and the present study confirms well known facts. A new contribution of this study would have been to present a rigorous and

systematic evaluation of the model sensitivity to soil profile thickness and initialization period. Unfortunately, the study falls short of delivering a new contribution due to a few important deficiencies in model boundary conditions as I explain below. I would suggest that the authors use appropriate boundary conditions and conduct new model sensitivity analyses that are scientifically defendable.

SPECIFIC COMMENTS

P3, L13-15. This paragraph seems to be out of place. I suggest deletion.

P4, L8. It is very unusual to have a 'grassland' ecosystem in a place like Normal Wells. I would suggest that the model be run with appropriate parameters to represent the vegetation typical of this environment.

P5, L4. The data from this borehole (84-1) is critically important for the evaluation of model performance. The model should be set up with the top boundary condition representing the vegetation characteristic of the local site, because the model simulation is compared against local data. The borehole data and site characteristics (including photographs) are publicly available from a report published by Natural Resources Canada. It appears that this site is located in a wetland surrounded by black spruce forests, typical of the Normal Wells region. To present a rigorous analysis (P19, L8), the model should use a set of parameters for wetlands, not grasslands for the top boundary.

P6, L22. Again, grass land cover is inappropriate for this particular model simulation.

P7, L1. The critical importance of geothermal heat flux applied to the bottom boundary is widely recognized by researchers in the permafrost modelling community, and is considered the standard practice. Geothermal heat flux data for the study region is readily available and used in previous studies (e.g. Zhang et al. 2008, cited by the authors). The absence of heat flux at the bottom boundary calls the scientific rigor of simulations in this study into question. I would strongly recommend that the authors

re-run the simulations using a proper bottom boundary condition. If the model cannot handle geothermal flux, then it is not an appropriate modelling platform for permafrost environments.

P8, L6. Please report mean annual air temperature and total precipitation for these years, preferably in a table format.

P9, L15. This statement is true with respect to annual temperature oscillation. However, the effects of lower-frequency temperature fluctuations (see Figure 6) can penetrate much deeper into the soil (see Figure 2). For a proper evaluation of model sensitivity, the non-oscillation depth should be defined using simulated temperature over multiple years.

P10, L9. It is not clear what is shown in Figure 5. The figure caption says it is annual average temperature, but it clearly is not. Please explain.

P14, L7. As I mentioned above (P9, L15), the temperature invariance in annual time scale does not necessarily indicate the insensitivity of the model to soil profile depth when lower-frequency fluctuations in atmospheric forcing are considered.

P16, L12-14. In addition to temperature, important variables in permafrost environments are the depth to the permafrost table (i.e. top of the permafrost) and the thickness of permafrost, as they exert strong influences on energy and water transfer processes. It is highly desirable to evaluate the model performance with respect to these key variables.

P19, L8. I cannot agree that this study presents a 'rigorous' analysis, as it suffers from fundamental problems concerning model boundary conditions. Please revise the boundary conditions and re-run the model simulations.

---

## Referee Comment (RC2) · Anonymous Referee #2 · 16 Jan 2018

The manuscript by Sapriza-Azuri and co-authors utilizes a well-established one-dimensional land-surface model (the Canadian Land Surface Scheme within the MESH) to establish 'how deep does the soil need to be' to appropriately model ground surface temperatures with permafrost to depth, and 'how long do we need' to initialize the climate for the simulation.

The main results from this paper are not new, although there are certainly some unique aspects to this paper. First, the fact that one needs very deep soil representation to account for permafrost is well known and unsurprisingly confirmed here. The second contribution of long time-scales of simulation is also not particularly novel, however the

authors have conducted the climate simulations in a relatively innovative manner by accounting for uncertainty and variability and providing robust estimates to this.

The paper as it stands requires revision to make an important scientific contribution. While a lot of good work has gone into this paper, it's unique contributions need to be highlighted. Furthermore, there needs to be proper accounting for the site selection and parametrization. It appears that the authors picked the data out of some publicly available archive and ran the simulation with little understanding of realistic boundary conditions. The authors need to carefully consider the surface conditions (vegetation, near surface soils) for this to be an appropriate and meaningful contribution. Norman Wells is not a grassland.

Specific Comments:

~P3 - line 13-15. This is out of place. Unsure as to why it is here. ~P3 - line 16-17 'there is no doubt that deeper soil....' Yes, this is well established. The question then is why is this work being completed? Additional referencing could be provided as to this. ~P3 - line 25 'the depth considered... generally arbitrary'. Can this statement be justified? I find it hard to believe that the work going in to establishment this depth is 'generally arbitrary'. Referencing would help. ~P3 - lines 30-34. I suggest the authors set up the paper less as a 'mystery' and with more direct language in how they are addressing the the questions in the paper. I find the set up very colloquial. ~p4 - line 8. The environment here is NOT characterized by grass. What is the influence of this on the simulation? Perhaps it is very little, but regardless, and appropriate upper boundary needs to be established here. ~P5, line 2 - The paragraph starts a bit awkwardly and there is no real justification as to WHY this site was chosen. There is historical data here, but there is elsewhere as well. ~the "Back to the past" language is again colloquial. I am not sure that this type of phrasing will be adopted in the scientific community and I would suggest the authors adjust their language to be one that is more technical. ~Figure 4 is nicely set up and I am wondering if Table 1 can be described in a more technical way or in a figure format as it is repetitive and as a reader not particularly

helpful. There is an obvious sequence here than can simply be described. ∼I am unsure as to how the parameters in Table 2 were given their upper or lower bounds. Yes, there was a Monte Carlo sampling with a uniform distribution, but LAI, minimum LAI, albedo, etc., to me seem as if they are incorrect for the environment. Please more carefully consider the rationale for this parametrization scheme and provide the reader with an understanding as to which one of these parameters is the most important for the setup and simulation. ∼P10, line 3. Please provide a reference to the end of the first sentence.

I have no real issue with the presentation of the results. As mentioned, a lot of thought and time went into the setup here and certainly a lot of computational resources were applied. I do, however, encourage the authors to highlight their scientific contribution here. The result of deeper soil configurations has been well defined for over a decade (or longer?) now. I believe that there is more value in exploring (appropriate) parameter sensitivity and the generation of relevant climate conditions. There were a lot of realizations here, but I am not sure that the authors have detailed the importance of these runs. What clear guidance can the authors provide other groups working in cold environments.

---

## Author Response (AR1)

**Response to Referee 1**

There are some minor changes in the response based on further updates of the manuscript.

**GENERAL COMMENTS**

The manuscript presents a sensitivity analysis of a Canadian one-dimensional land surface model, MESH to the thickness of modeled soil profile and the length of model initialization period. The main conclusions are that a soil profile of 20m or greater is necessary for this particular model to represent the energy dynamics of permafrost, and that an initialization period much longer than 100 years is necessary to condition the model properly. The same results have been reported by a number of previous researchers using different permafrost models, and the present study confirms well known facts. A new contribution of this study would have been to present a rigorous and systematic evaluation of the model sensitivity to soil profile thickness and initialization period. Unfortunately, the study falls short of delivering a new contribution due to a few important deficiencies in model boundary conditions as I explain below. I would suggest that the authors use appropriate boundary conditions and conduct new model sensitivity analyses that are scientifically defendable.

**Response to General Comments**

*We thank the referee for his/her insightful comments. While we certainly agree with the referee on the significance of geothermal flux, we would point out that the difference between the common implementation of the current generation of Land Surface Schemes (LSS), applied as the lower boundary condition for regional/general circulation models and hydrological applications, and that of permafrost models used to predict and evaluate the evolution of permafrost. In the former, the geothermal flux is commonly ignored in the literature and most of existing models do not have the parameterization to include it (the common assumption is no heat flux at the bottom of the soil layers), while in the latter, geothermal flux is considered an essential component of modelling. In response to the referee's comment, we have extended the analysis by including a new set of simulations with a constant geothermal flux (0.083 W/m2) at the bottom, based on available measurements at Norman Wells (Garland, G.D. and Lennox, D.H., Heat Flow in Western Canada. Geophys. J.R. Astron. Soc., 6,245-262, 1962).We have run the same set of 50 parameters with 17 different soil configuration combinations (850 simulations) for the average climate year (1945). Figure 1 shows the results obtained. Although some small differences are observed, the conclusions remain the same (e.g., 20 meters of soil depth are needed). The main difference is seen in some soil configurations having slightly warmer soil profile when "ggeo" flux is included. Figure 2 shows the cumulative distribution function of the difference of soil temperature at the non-oscillation depth with and without ggeo flux at the bottom. Approximately, in 60% of the simulation the difference is within +/- 0.15◦C. Upper boundary condition. We have corrected the description of the site location and clarified the assumptions made about the upper boundary condition. The land cover in the manuscript has been corrected to be a composition of moss lichen groundcover, ericaceous shrubs, black spruce and tamarack trees in an open canopy density (Smith et al., 2004). In this study, we perturbed the canopy parameters by a Monte Carlo analysis, not using a specific land cover type based on a look-up table. The range of variation selected in such a way that it covers most of the possible land covers present in the area. The purpose of that was to analyse the uncertainty in parameter values on the definition of the soil configuration and in land surface schemes that are typically run at a grid size ranging from ~10*10 to ~250*250 km2 (which can be different from an analysis performed to represent the processes at a point). We have*

*included two new sections to describe the boundary conditions used and to show the results obtained of the geothermal flux at the bottom. To better define the scope of our work we have restructured the introduction to better reflect the novelty of our contribution.*

**SPECIFIC COMMENTS**

1. P3, L13-15. This paragraph seems to be out of place. I suggest deletion.

   *Thank for your suggestion, we have removed the paragraph.*

2. P4, L8. It is very unusual to have a 'grassland' ecosystem in a place like Normal Wells. I would suggest that the model be run with appropriate parameters to represent the vegetation typical of this environment.

   *We agree with the referee. The confusion here it is was derived from the Land Cover map used in this analysis, that came from a reclassification of a land cover map from a bigger area for the Mackenzie basin, where shrubs, grass and other cover were grouped together in a single unit unfortunately named grassland . In addition also the original pixels were upscaled and we only pick the dominant land cover type. However, in really, as the canopy parameters were perturbed by a Monte Carlo analysis, in fact, we have not used a specific land cover type based on a look-up table. The range of variation cover most of the possible land cover present in the area. The purpose to do not attach to a specific set of parameters representing a land cover was to show that regardless of their value you need to have a deeper soil configuration in cold regions. To avoid any kind misunderstanding we have corrected the land cover specific for the place adding a complete description of the site vegetation and canopy based on the site description reported in Smith et al., (2004). We removed any grass land cover reference from the text. As explained, we considerate that re-run the simulation are not necessary.*

3. P5, L4. The data from this borehole (84-1) is critically important for the evaluation of model performance. The model should be set up with the top boundary condition representing the vegetation characteristic of the local site, because the model simulation is compared against local data. The borehole data and site characteristics (including photographs) are publicly available from a report published by Natural Resources Canada. It appears that this site is located in a wetland surrounded by black spruce forests, typical of the Normal Wells region. To present a rigorous analysis (P19, L8), the model should use a set of parameters for wetlands, not grasslands for the top boundary.

   *Please see response to comment #2.*

4. P6, L22. Again, grass land cover is inappropriate for this particular model simulation.

   *Please see response to comment #2.*

5. P7, L1. The critical importance of geothermal heat flux applied to the bottom boundary is widely recognized by researchers in the permafrost modeling community, and is considered the standard practice. Geothermal heat flux data for the study region is readily available and used in previous studies (e.g. Zhang et al. 2008, cited by the authors). The absence of heat flux at the bottom boundary calls the scientific rigor of simulations in this study into question. I would strongly recommend that the

authors re-run the simulations using a proper bottom boundary condition. If the model cannot handle geothermal flux, then it is not an appropriate modeling platform for permafrost environments.

*We understand and also share your point. As we respond to the general comments, the focus of the present analysis was more to address the kind of Land Surface Schemes commonly used as lower boundary condition for regional/global circulation models and hydrological applications, rather than the kind of permafrost models used to predict and evaluate the evolution of permafrost presence. In this kind of application there is almost no inclusion of geothermal flux (model do not have the parameterization) and the common assumption is no heat flux. Class allow to include a constant geothermal flux at the bottom, we have included a sub set of simulation to compare the effect of geothermal flux as was described in the response to general comments.*

6.  P8, L6. Please report mean annual air temperature and total precipitation for these years, preferably in a table format.

    *Added in Table 2.*

7.  7. P9, L15. This statement is true with respect to annual temperature oscillation. However, the effects of lower-frequency temperature fluctuations (see Figure 6) can penetrate much deeper into the soil (see Figure 2). For a proper evaluation of model sensitivity, the non-oscillation depth should be defined using simulated temperature over multiple years.

    *Thanks for the comment. However, in the Experiment 1 we are running in a spinup mode recycling the same year over 2000 times. After that cycling we assume that a quasi-equilibrium between climate condition and the ground thermal state was reached for a year. One of the things that we are trying to show here is the effect on the selection of climate condition to stabilize a model, so only one year is used.*

8.  8. P10, L9. It is not clear what is shown in Figure 5. The figure caption says it is annual average temperature, but it clearly is not. Please explain.

    *We apology for the confusion here, maybe the selection of words were not the best. The label: "... Trend comparison of annual average air temperature with subtracted mean for the whole period ..." have been modified to ". . .Trend comparison of residual of the difference between annual average air temperature and...".*

9.  P14, L7. As I mentioned above (P9, L15), the temperature invariance in annual time scale does not necessarily indicate the insensitivity of the model to soil profile depth when lower-frequency fluctuations in atmospheric forcing are considered.

    *Please see response to comment #7 P9 L15.*

10. P16, L12-14. In addition to temperature, important variables in permafrost environments are the depth to the permafrost table (i.e. top of the permafrost) and the thickness of permafrost, as they exert strong influences on energy and water transfer processes. It is highly desirable to evaluate the model performance with respect to these key variables.

    *Thanks for the comment. However, as we responded to comment 5, the kind of model that we are addressing in the manuscript are more related to the common Land*

*Surface Model used in Regional/Global Circulation and hydrology models. It is out of the scope of the paper to have a complete and exhaustive permafrost simulation. Finally, we try to keep it simple, the analysis already have huge number of comparison, and we prefer to maintain the selected variables showed.*

11. P19, L8. I cannot agree that this study presents a 'rigorous' analysis, as it suffers from fundamental problems concerning model boundary conditions. Please revise the boundary conditions and re-run the model simulations.

*We appreciate your comment. Of course that word 'rigorous' has some implications, however, we have jointly cover many source of uncertainty not analyzed before, that could affect the definition of the depth of the soil configuration and how is initialized. Again, as we pointed out in response to comment #5 and #10 the scope of the paper is in other kind of models.*

*Figures*

[Figure]

Figure 1 *2d-Histogram of SCD and $h_T$–non-oscillation depth. Counts are normalized by the number of simulation by SCD. The black line represents the limit to reach or not the $h_T$–non-oscillation conditions. Bins to the left represent SCDs that never reach the $h_T$–non-oscillation condition. a) No Geothermal flux, b) Constant Geothermal flux as lower boundary condition at the bottom of the soil layers.*

[Figure]

*Figure 2. Cumulative distribution function (CDF) of the soil temperature difference at the hT-non-oscillation depth between simulations with and without geothermal flux.*

**Response to Referee 2**

There are some minor changes in the response based on further updates of the manuscript.

**GENERAL COMMENTS**

The manuscript by Sapriza-Azuri and co-authors utilizes a well-established one dimensional land-surface model (the Canadian Land Surface Scheme within the MESH) to establish 'how deep does the soil need to be' to appropriately model ground surface temperatures with permafrost to depth, and 'how long do we need' to initialize the climate for the simulation The main results from this paper are not new, although there are certainly some unique aspects to this paper. First, the fact that one needs very deep soil representation to account for permafrost is well known and unsurprisingly confirmed here. The second contribution of long time-scales of simulation is also not particularly novel, however the authors have conducted the climate simulations in a relatively innovative manner by accounting for uncertainty and variability and providing robust estimates to this. The paper as it stands requires revision to make an important scientific contribution. While a lot of good work has gone into this paper, it's unique contributions need to be highlighted. Furthermore, there needs to be proper accounting for the site selection and parametrization. It appears that the authors picked the data out of some publicly available archive and ran the simulation with little understanding of realistic boundary conditions. The authors need to carefully consider the surface conditions (vegetation, near surface soils) for this to be an appropriate and meaningful contribution. Norman Wells is not a grassland.

**Response to General Comments:**

*We thank the referee by the constructive nature of his/her criticisms. To highlight the novelty of our contribution we have restructured the introduction and added more references. We have separated the literature review of previous works into three parts (1) the need for deep soil configuration, (2) the need for proper initialization, including the incorporation of uncertainty and the reconstruction of past climate time series, and (3) parameter uncertainty.*

*We pointed out in the manuscript the following lines:*

*"…Despite significant advances, as briefly outlined above, the appropriate soil configuration depth (SCD) in land surface modelling of cold regions remains an open question. This question is further complicated by the fact that parameter uncertainty is typically ignored in LSMs, and parameter values are usually collected from look-up tables based on land cover and soil maps (Mendoza et al., 2015). Related to this, there have been some previous efforts for "sensitivity analysis" of model outputs to parameters (Razavi and Gupta, 2015) but these have been mainly limited to comparisons of different cover types (e.g., Paquin and Sushama, 2015; Yang et al., 1995) with some few exceptions (e.g., Bastidas et al., 2006).*

*In this paper, we focus on the three inter-related aspects of LSMs, namely soil depth, parameter uncertainty, and initializations, together to address the above question. Unlike the previous studies that focus on each aspect in isolation, this study looks at their joint and individual effects. We set up a series of systematic modelling experiments with the following three objectives to (1) identify the appropriate SCD for a given LSM and location in the presence of uncertainty in model parameter values and climate conditions, (2) assess the significance of including/excluding geothermal flux as the lower boundary condition in an LSM, (3) develop an*

*initialization procedure for LSMs in cold regions based on paleo-reconstructions of climate variables and statistical bootstrapping. …"*

*The list of reference added:*
- *Yang, Z.-L., R. E. Dickinson, A. Henderson-Sellers, and A. J. Pitman. Preliminary study of spin-up processes in land surface models with the first stage data of Project for Intercomparison of Land Surface Parameterization Schemes Phase 1(a), J. Geophys. Res., 100(D8), 16553–16578, doi:10.1029/95JD01076, 1995.*
- *Rodell, M., P.R. Houser, A.A. Berg, and J.S. Famiglietti, Evaluation of 10 Methods for Initializing a Land Surface Model. J. Hydrometeor., 6, 146–155, https://doi.org/10.1175/JHM414.1, 2005. Shrestha, R., and P. Houser, A heterogeneous land surface model initialization study, J. Geophys. Res., 115, D19111, doi:10.1029/2009JD013252, 2010.*
- *Mendoza, P. A., M. P. Clark, M. Barlage, B. Rajagopalan, L. Samaniego, G. Abramowitz, and H. Gupta. Are we unnecessarily constraining the agility of complex process-based models?, Water Resour. Res., 51, 716–728, doi:10.1002/2014WR015820, 2015.*
- *Bastidas, L. A., T. S. Hogue, S. Sorooshian, H. V. Gupta, and W. J. Shuttleworth, Parameter sensitivity analysis for different complexity land surface models using multicriteria methods, J. Geophys. Res., 111, D20101, doi:10.1029/2005JD006377, 2006.*
- *Razavi, S., and H. V. Gupta, What do we mean by sensitivity analysis? The need for comprehensive characterization of ''global'' sensitivity in Earth and Environmental systems models, Water Resour. Res., 51, 3070–3092, doi:10.1002/2014WR016527, 2015.*

*Regarding to surface conditions, please refer to response to specific comment #5.*

**Specific Comments**

1. P3 - line 13-15. This is out of place. Unsure as to why it is here.

   *Thank for your suggestion, we have removed the paragraph.*

2. P3 - line 16-17 there is no doubt that deeper soil....' Yes, this is well established. The question then is why is this work being completed? Additional referencing could be provided as to this.

   *We appreciate your comment. We have removed those lines from the text and restructured the introduction and adding references. These change look for a better definition of the scope and to highlight the new contribution. Please refer to response to general comments to the reference added and main change in the introduction.*

3. P3 - line 25 'the depth considered... generally arbitrary'. Can this statement be justified? I find it hard to believe that the work going in to establishment this depth is 'generally arbitrary'. Referencing would help.

   *We have removed that line from the text. Please refer to response to general comments to the reference added and main change in the introduction.*

4. P3 - lines 30-34. I suggest the authors set up the paper less as a 'mystery' and with more direct language in how they are addressing the questions in the paper. I find the set up very colloquial.

*Thanks for the suggestion. Those lines were removed from the texts and the introduction restructured. Please refer to response to general comments to the reference added and main change in the introduction.*

5. P4 -line 8. The environment here is NOT characterized by grass. What is the influence of this on the simulation? Perhaps it is very little, but regardless, and appropriate upper boundary needs to be established here.

*We agree with the referee and regret that the landcover was misrepresented in the original manuscript. Having a large-scale modelling approach in mind, the dominant landcover in a pixel of 10*10 km2 was named grassland. The confusion here was due to the Land Cover map used in this analysis that came from a reclassification of a land cover map from a bigger area for the Mackenzie basin, where shrubs, grass and other types of land covers were grouped together in a single unit, unfortunately named grassland. In addition, the original pixels were upscaled and only the dominant land cover type was picked. We fixed this problem in the writing of the revised manuscript. We have corrected the land cover type of this specific location and added a complete description of its vegetation and canopy based on the site description reported in Smith et al., (2004). The analyses and results didn't need to be changed; the reason is as the canopy parameters were perturbed by a Monte Carlo analysis, we have not used a specific land cover type based on a look-up table. The range of variation covered most of possible land cover types present in the area. As an aside, we mention that our analyses showed that regardless of parameter values, a deep soil configuration would be needed in large-scale modelling of cold regions.*

6. P5, line 2 - The paragraph starts a bit awkwardly and there is no real justification as to WHY this site was chosen. There is historical data here, but there is elsewhere as well.

*We have changed the start of the paragraph as follow: "Annual soil temperature profiles are available based on the maximum and minimum daily average of soil temperature at several borehole locations in the Mackenzie Valley, administrated by the Geological Survey of Canada (Smith et al., 2004). . . ." The selection was made on the availability of data and, of course other places could be selected. As future work the plan is to generalize to other locations as was pointed out in the conclusion.*

7. the "Back to the past" language is again colloquial. I am not sure that this type of phrasing will be adopted in the scientific community and I would suggest the authors adjust their language to be one that is more technical.

*Thanks for the suggestion. We have changed 'Back to the past" to "Paleo-Reconstruction".*

8. Figure 4 is nicely set up and I am wondering if Table 1 can be described in a more technical way or in a figure format as it is repetitive and as a reader not particularly helpful. There is an obvious sequence here than can simply be described.

*We have changed Table 1 to a figure format. The Figure 5 has the model discretization.*

9. I am unsure as to how the parameters in Table 2 were given their upper or lower bounds.

*Yes, there was a Monte Carlo sampling with a uniform distribution, but LAI, minimum LAI, albedo, etc., to me seem as if they are incorrect for the environment. Please more carefully consider the rationale for this parametrization scheme and provide the reader with an understanding as to which one of these parameters is the most important for the setup and simulation. The rational here was to have more flexibility in the parameter range so, the result could be more robust about of what does matter in norther places (climate, soils depth or parameters). The parameter range cover mainly most of the land cover presence in that area from. To clarify this point we have added the following lines. . . ". . . The range of the canopy parameters values represent different vegetation cover that are present in the area based on the look-up table from CLASS user manual (Versegey, 2009). ..."*

10. P10, line 3. Please provide a reference to the end of the first sentence

*We have added the following reference: Yang, Z.-L., R. E. Dickinson, A. Henderson-Sellers, and A. J. Pitman (1995), Preliminary study of spin-up processes in land surface models with the first stage data of Project for Intercomparison of Land Surface Parameterization Schemes Phase 1(a), J. Geophys. Res., 100(D8), 16553–16578, doi:10.1029/95JD01076.*

11. I have no real issue with the presentation of the results. As mentioned, a lot of thought and time went into the setup here and certainly a lot of computational resources were applied. I do, however, encourage the authors to highlight their scientific contribution here. The result of deeper soil configurations has been well defined for over a decade (or longer?) now. I believe that there is more value in exploring (appropriate) parameter sensitivity and the generation of relevant climate conditions. There were a lot of realizations here, but I am not sure that the authors have detailed the importance of these runs. What clear guidance can the authors provide other groups working in cold environments.

*We appreciate your comment. We have restructured the introduction in a way to better highlight the main contribution of this work. Please see response to general comments. We have added to the Discussion and conclusion section the following lines in relation to reconstruction of past climate time series: ". . . . An important remark here is that the effect of stochasticity in the reconstructed time series is minimal, so what is important is to reproduce historical (low frequency) trends. ..." The recommendation are detailed in the Discussion and Conclusion section and they are: (a) Minimum soil depth of 20 m (b) Initialization in two stages: i. First stage spin-up using a single average year to reach quasiequilibrium condition on fluxes and state variables. ii. Reconstruction of past climate time series, to allow the model evolve over time on the time period preceding the period of records as to be able to simulate current conditions. iii. Recognize the parameter uncertainty.*

**List of main change in the manuscript:**

- Last paragraphs of the Introduction to better reflex the objective and the novelty of the paper.
- Correction of the description of the land cover
- Table 1 represented as Figure 5
- Table 1 added that describe the temp and precip for each climate year
- New section describing the lower boundary conditions
- We include in the analysis the incorporation of ggeo flux as lower boundary condition at bottom of the soil layers
- New result section that describes the effect of include or not ggeo flux. Two new figures that show those results.
- Figures from result section -Initialization by Paleo-Reconstructions that compare SCD and RMS were grouped in one figure (Figure 15a,b)
- Update in conclusion including new results.

[revised manuscript text omitted]
 untilup to the point that stabilization insoil temperature profile is stabilizedreached. Then, in the second step, we recommend usinggenerating multi-century long time series of climate variables generated records based on the procedure proposed in this study. The proposed

30 procedure reconstructspaleo-reconstructions, and running the model of step 1 on that. This will let the model evolve over time on the time seriesperiod preceding the period of temperature records as to be able to simulate current conditions. Here, we reconstructed past climate using proxy recordsdata of summer temperatures derived from tree rings and generates the concurrent time series of other climate variables such as precipitation by, applying block -bootstrapping on historical records.

We were able to reproduce quite well the past trends of summer temperature and we included the effect of uncertainty in the

climate time series by generating 100 realizations. An important remark here is that the effect of short-time scale (e.g., annual) fluctuations in the reconstructed time series used for initialization was minimal, while low frequency trends were important. The length of reconstructions required for proper initialization is longer for deeper SCDs. The number of years that are necessary to go back to the past will be a function of how deep is the SCD chosen. Deeper SCDs retain more memory of past

5  climate and require longer spin up periods.

Finally, we envision our future work being directed to generalize the results obtained here by extending the analyses to other locations places where observations (of past climate and soil profile temperature and past climate ) are available. Furthermore, implementing a variable SCD , increasing the number of parameters sampled to better explore the parameter space, and comparing several model parametrizations, including the effect of heat flux as a lower boundary condition at the bottom of the

10  soil. For application in regional and global models may, the SCD can be investigated, variable as also was proposed by Brunke et al. (2016) for in the Community Land Model version 4.5. However, the 
[revised manuscript text omitted]